# Beneficial Bacteria Isolated from Food in Relation to the Next Generation of Probiotics

**DOI:** 10.3390/microorganisms11071714

**Published:** 2023-06-30

**Authors:** Barbara Sionek, Aleksandra Szydłowska, Dorota Zielińska, Katarzyna Neffe-Skocińska, Danuta Kołożyn-Krajewska

**Affiliations:** 1Department of Food Gastronomy and Food Hygiene, Institute of Human Nutrition Sciences, Warsaw University of Life Sciences (WULS), Nowoursynowska St. 159C, 02-776 Warszawa, Poland; aleksandra_szydlowska@sggw.edu.pl (A.S.); dorota_zielinska@sggw.edu.pl (D.Z.); katarzyna_neffe_skocinska@sggw.edu.pl (K.N.-S.); 2Department of Dietetics and Food Studies, Faculty of Science and Technology, Jan Dlugosz University in Czestochowa, Al. Armii Krajowej 13/15, 42-200 Częstochowa, Poland

**Keywords:** probiotics, NGPs, acetic acid bacteria (AAB), gut microbiome, human health, functional food

## Abstract

Recently, probiotics are increasingly being used for human health. So far, only lactic acid bacteria isolated from the human gastrointestinal tract were recommended for human use as probiotics. However, more authors suggest that probiotics can be also isolated from unconventional sources, such as fermented food products of animal and plant origin. Traditional fermented products are a rich source of microorganisms, some of which may have probiotic properties. A novel category of recently isolated microorganisms with great potential of health benefits are next-generation probiotics (NGPs). In this review, general information of some “beneficial microbes”, including NGPs and acetic acid bacteria, were presented as well as essential mechanisms and microbe host interactions. Many reports showed that NGP selected strains and probiotics from unconventional sources exhibit positive properties when it comes to human health (i.e., they have a positive effect on metabolic, human gastrointestinal, neurological, cardiovascular, and immune system diseases). Here we also briefly present the current regulatory framework and requirements that should be followed to introduce new microorganisms for human use. The term “probiotic” as used herein is not limited to conventional probiotics. Innovation will undoubtedly result in the isolation of potential probiotics from new sources with fascinating new health advantages and hitherto unforeseen functionalities.

## 1. Introduction

The basic concern of man has always been and still is the problem of obtaining food, only then did they strive for its quality. In societies where obtaining food is not a problem, there is a desire to have such products that meet the sophisticated needs of consumers. Food should be tasty, absolutely safe for health, of appropriate quality, and perhaps it should also fulfill other functions?

This is the direction that scientists, technologists, and nutritionists are following. Is it possible to produce products that, apart from nutritional functions, will also fulfill other, primarily health, tasks? In this way, the concept was born and products that are included in the group called functional foods were created.

In recent years, so-called probiotic food has become a fast-developing segment in the functional food market. It contains live colonies of intestinal bacteria, which, through the ability to colonize the digestive tract, affect various processes of metabolism of the body, improve human health, or cure diseases.

The way in which probiotics are defined changes with the development of knowledge about them. According to International Scientific Association for Probiotics and Prebiotics (ISAPP) consensus, the term “probiotic” may be used to refer to many types of microorganisms that demonstrate health benefits for the host, while remaining alive [1]. Metabolites, as well as dead cells of microorganisms, were excluded from the definition of a “probiotic”. Additionally, it was agreed that “probiotics” are not undefined consortia of microorganisms (such as fecal microbiota transplants) or fermented foods containing undefined microorganisms [2].

Zielińska and Kołożyn-Krajewska (2018), in a previous publication, concluded that probiotic microorganisms, in addition to the conventional source (healthy human digestive tract), may come from unconventional sources, such as the digestive tract of animals, breast milk, food (fermented and unfermented), air, or soil [2]. The isolation, identification, and assessment of safety and probiotic properties of new, “wild” strains of microorganisms from traditional food is a necessary practice, especially for the development of technology for the production of vaccines dedicated to food. New vaccines, in addition to protective properties (bacteriostatic and bactericidal), may bring additional values related to the improvement of consumer health.

Extending the term “probiotic” to include bacteria isolated from traditionally and spontaneously fermented foods seems justified. Microorganisms isolated from fermented products constitute the microbiota of the environment in which the products were produced. If they are tested, particularly in terms of their probiotic properties and safety, they may constitute an interesting alternative to gut bacteria [2].

Recently, advanced genetic sequencing tools opened up new perspectives of knowledge of the gut microbiota composition. A group of microorganisms called next-generation probiotics (NGPs) was selected [3]. They attracted the interest of scientists, mainly due to their potential human health benefits. The list of scientific reports on the importance of NGPs for maintaining the balance of the human microbiome and their impact on disease prevention and beneficial therapeutic effects is growing. Nevertheless, of the gathered knowledge of the microbiome interaction with a human host, and of the role of gut dysbiosis, and acute and chronic inflammation-related diseases, the expected use of NGPS in nutritional or medicinal products is limited by insufficient safety assessments. It requires a series of randomized trials, which are, until now, scarce.

Apart from that, it has also been shown that some of the beneficial properties of probiotics may be due to the substances secreted by them, products of their metabolism, or substances released as a result of the lysis of their cells. To define these substances, terms such as postbiotics, metabiotics, bacterial metabolites, cell-free supernatants, or metabolic residues of probiotic activity are used, as well as paraprobiotics, non-viable probiotics, inactivated probiotics, or ghost probiotics [4].

Conducted experiments, research, and extensive analyses expand our knowledge on understanding the role of “good” microorganisms in shaping human health. It seems that in the future, the term “probiotic” will have a broader meaning. The aim of the presented review is to demonstrate and discuss the issues related to beneficial bacteria isolated from food, and the group of next-generation probiotics, recently isolated from the human microbiome.

## 2. The Potential of Probiotic Microorganisms Isolated from Food

### 2.1. Probiotics—Revisited Definition and Criteria of Probiotic Status

The Russian Nobel laureate Élie Metchnikoff, who promoted the idea that altering the composition of the intestinal microbiome with specific beneficial bacteria has the potential to improve health and lengthen life, coined the term “probiotic” [5].

Currently, probiotic microorganisms are defined as “live microorganisms that, when administered in adequate amounts, confer a health benefit on the host”. This definition is inclusive of a broad range of microbes and applications, whilst capturing the essence of probiotics (microbial, viable, and beneficial to health) [6].

In agreement with this definition was the International Scientific Association of Probiotics and Prebiotics (ISAPP) in a position statement [1]. Defining the criteria for probiotics was extremely significant for all parties concerned.

Even so, while the term “probiotic” is used widely in both food and supplement categories, it is often misleading. A candidate probiotic strain should meet certain imposed criteria such as: being characterized safe for intended use in at least one positive relevant human study, and a certainty that the product delivers the studied dose and remains viable until the end of the shelf life.

Clarification of the minimal standards required for the correct usage of the term “probiotic” is especially helpful now that new “biotic” terminology, such as pharmabiotic, psychobiotic, postbiotic, synbiotic, and others are entering the global lexicon [7].

The term “probiotic”, as used herein, is not limited to conventional probiotics. Innovation will undoubtedly result in the isolation of potential probiotics from new sources with fascinating new health advantages and hitherto unforeseen functionalities. This definition does not exclude these so-called “next generation probiotics”, which, in some situations, could be thought of as live biotherapeutics. However, depending on the intended use, appropriate safety, legal, and ethical issues must be addressed in the development of such probiotics. For example, where the Nagoya protocol is applicable, it must be followed, and when removing microbes from people, informed consent must be obtained [8].

Lactic acid bacterial (LAB) strains are members of our intestinal microbiota and widely being used as probiotics [9]. Probiotic microorganisms are mainly *Lactobacillus* and *Bifidobacterium*, but *Lactococcus*, *Streptococcus*, *Enterococcus*, *Propionibacterium*, and *Saccharomyces* yeasts are also important and widely existing [10]. The first probiotics regarding technological properties and their use in the food industry were *Lactobacilli* and *Bifidobacteria* [4]. Some authors are also pointing out the need for diagnosing potential probiotic properties of selected strains of acetic acid bacteria (AAB) [11,12].

### 2.2. Isolation Sources of Probiotic Microorganisms

According to FAO/WHO, the human gastrointestinal tract (GIT) gives rise to the conventional source of probiotics for human use [13].

Some authors claim that it is justified to expand the definition of “probiotic” to include microorganisms extracted from traditionally fermented food. The microbiota of the environment in which the products were created is represented by bacteria that have been isolated from fermented products. They could make an intriguing alternative to gut bacteria if studied, particularly in terms of their probiotic characteristics and safety [2].

Isolating probiotic microorganisms from food is essential for the development of new probiotic products and for improving consumer health. The probiotic potential of microorganisms isolated from food depends on many factors, such as their ability to survive in the gastrointestinal tract, their ability to colonize the intestinal tract, and their ability to produce metabolites that benefit health [14].

In Figure 1 the isolation sources of probiotic microorganisms in food are shown.

The probiotics used in humans commonly come from dairy foods, whereas the sources of probiotics used in animals are often the animals’ own digestive tracts [15].

Select probiotics from so-called “unconventional sources” are anticipated to become more popular. To prevent dairy consumption in those who are lactose intolerant, alternate sources of probiotic selection have become more popular. Unusual sources of microorganisms, such as non-intestinal sources, non-dairy fermented (raw fermented meat products, fermented fish and seafood, and pickled vegetables), and unfermented food products (fruits and vegetables), soil, and air, were screened for potential probiotics [16,17,18,19,20].

Some authors also report a strong need for new bacterial strains to be isolated from the wild to increase the genetic diversity of the collection of LAB strains intended for novel product technology [21,22].

### 2.3. Industrial Use of Probiotic Lactic Acid Bacterial Strains

The LAB strains demonstrate the ability to produce lactic acid as the main end-product of their anaerobic metabolism and the ability to synthesize a wide range of metabolites that beneficially affect the sensorial, nutritional, and technological properties of fermented foodstuffs. In connection with the properties shown, they are used as probiotics, starter cultures in the food industry, and in the production of some nutraceuticals [23,24].

For LAB strains to be employed as starter cultures, they must be non-pathogenic, probiotic, and technologically adapted to the fermentation environment. According to Bonatsou et al. (2017) [25], LAB strains are known for their probiotic characteristics, and their ability to produce antibacterial chemicals, sugar polymers, sweeteners, aromatic compounds, vitamins, or useful enzymes.

The European Union’s research projects on lactic acid bacteria are supported by the Lactic Acid Bacteria Industrial Platform (LABIP), an industrial platform. The European Economic Association LABIP was established in 1994. Companies that utilize or manufacture LAB strains and have facilities for production or research inside the EU are members of LABIP. The coordination of communication regarding issues of industrial importance between academia, industry, and EU authorities is one of LABIP’s goals. The expert workshop “future access and improvement of industrial LAB cultures” was organized and sponsored by LABIP [26].

In addition, our Department of Food Gastronomy and Food Hygiene, Institute of Human Nutrition Sciences at Warsaw University of Life Sciences is in possession of its own collection of over 200 pure cultures of *Lactobacillus* sp. and other lactic acid bacteria isolated from spontaneously fermented food products.

### 2.4. Potential Benefits of Probiotics

The species belonging to the order *Lactobacillales* are abundant in nature and thus suitable for gut microbiota modulation and incorporation into many food systems [27].

Due to the development and progress of science, the issue of the importance of the human microbiome on the overall condition of the human body is now becoming increasingly known [28]. It is worth mentioning that when consumed as a monoculture or mixed culture, probiotics affect the host by enhancing the quality of the indigenous microflora in the gastrointestinal tract [29].

The mechanisms of action of microorganisms with probiotic properties, thanks to which they affect human health, are not fully explained but may include competitiveness in relation to intestinal pathogens, the decomposition of carcinogens and antinutritional factors, the production of antimicrobial metabolites, and the modulation of the immune response of the gastrointestinal mucosa. Probiotics used to support the role of intestinal microorganisms are currently not only important to support resistance against infections but also can be genetically modified to obtain strains with targeted therapeutic and technological action. Therefore, the safety of their use and further research on their effectiveness and stability during their use in both pharmaceutical microbiology and the development of recipes for new products is extremely important [30].

The Functional Food Center (FFC) defines “functional foods” as “Natural or processed foods that contain biologically-active compounds; which, in defined, effective, non-toxic amounts, provide a clinically proven and documented health benefit utilizing specific biomarkers, to promote optimal health and reduce the risk of chronic/viral diseases and manage their symptoms”. This definition highlights the positive effect of bioactive ingredients on physiological mechanisms that are important for maintaining human health [31].

While there is evidence supporting the potential health benefits of probiotics, the mechanisms of action and clinical efficacy of specific strains or combinations of strains are still being researched [32,33,34].

Recent studies have shown the beneficial effect of probiotic microorganisms on human health, including the immune system [35,36], nervous system [37,38], and their biological activities for the prevention of metabolic diseases [39,40]. Probiotics have been shown to have positive effects on health in the treatment of a wide range of illnesses, including inflammatory bowel disease, irritable bowel syndrome, constipation, antibiotic-associated and acute diarrhea, allergy-related problems, hypertension, and diabetes. However, the selection of appropriate bacterial strains and the conditions for their breeding and development of functional food products with probiotics is key to achieving optimal health effects. The COVID-19 epidemic has increased interest in the effects of food on the resistance of the human body (especially in the absence of the vaccine). Furthermore, after vaccination, food was treated as a natural source of immunomodulation [41].

Due to their demonstrated efficacy in reversing the pathogenicity of food-borne pathogens, probiotic bacteria potentially constitute a possible alternative in the prevention and control of food-borne diseases [5].

The formation of hydrogen peroxide, lactic acid, bacteriocin-like compounds, immune system stimulation, and the regulation of intestinal microbiota are a few of the methods through which LAB strains appear to be active against bacterial infections [42,43]. Additionally, LAB strains can inhibit pathogen adherence by competing with them for the binding sites on intestinal epithelial cells, which reduces colonization and delays the onset of infection [44,45].

In summary, there are many bacterial strains that can be used as probiotics. However, selecting the appropriate strains that exhibit desirable characteristics, such as acid and bile resistance, adherence to the gut epithelium, and the ability to produce beneficial metabolites, can be complex. Moreover, characterizing the strains for safety and stability is crucial to ensure their suitability for use as probiotics.

## 3. Insight in Probiotic Diversity: Conventional and Unconventional Sources of Probiotics

The isolation of potential probiotic strains is the first step included in FAO/WHO (2001) guidance. Scientists agree that we still need new, well-studied strains of bacteria that could be used as probiotics, especially for specific treatment use [46].

Traditionally, probiotics are selected among bacteria of the genus of *Lactobacillus* and *Bifidobacterium.* It was claimed that probiotics intended for human use should be from “human or food origin” because those strains are more likely to be safe for humans and able to attach to human intestinal epithelial cells. Conventional sources of potential probiotic strains isolation are the gastro-intestinal tract and the breast milk. It is well known that human milk is an important factor in the colonization of the aseptic intestine of a newborn with the first microbiota. Based on this, it has been suggested that human milk contains bacterial strains with the potential to be used as probiotic agents. In addition, the feces of adults, children, and infants have also been found to be the potential best source of probiotics for human use, because of their possibility to withstand gastrointestinal transit and colonize the intestines for beneficial actions [47].

Moreover, animal-origin food sources (such as milk, meat, and honey), as well as plant-origin foods (fermented and non-fermented), could be alternative sources for the isolation of potential probiotic strains. This concept assumes that many of the well-studied strains isolated from feces are not of human origin (e.g., *Bifdobacterium animalis* subsp. lactis and *Saccharomyces cerevisiae* var. *boulardii*). Therefore, it is suggested that food is the primary source of microorganisms present in the gastrointestinal tract. Many studies have shown that probiotic strains isolated from traditional and regional fermented foods can be used for the development of starter cultures for the commercial production of fermented probiotic foods [2].

The new concept of probiotic use is “personalized” or “precision” treatment, which includes the specific characteristics of the host, thus avoiding the “one size fits all” approach, which has often proven ineffective. A personalized strategy using probiotic therapy was recently proposed and proved in the animal model study for intestinal diseases, where commensal bacteria isolated from the gut microbiota of a healthy host were stored in a ‘microbiota biobank’, and after specific characteristics were tested, the bacteria were successfully used as a therapy for dysbiosis-related disease [48]. This method hypothesizes that commensal bacteria isolated in a personalized way would facilitate colonization based on the specific genetic makeup of the host.

## 4. Concept of Next-Generation Probiotics (NGPs)

The human gut microbiome is a huge and unexplored ecosystem of microbes. It is estimated that microbes in the human intestine can reach up to ten trillion cells (approximately ten times more than human cells) [49]. Seven divisions of bacteria (*Firmicutes*, *Bacteroidetes*, *Actinobacteria*, *Fusobacteria*, *Proteobacteria*, *Verrucomicrobia*, and *Cyanobacteria*) belong to the microbial community of the intestine [50]. Approximately 80% of microbes in the intestine ecosystem are still unknown [51]. Some of these microbes are potentially beneficial for the host and may be considered as next-generation probiotics. The comparative analysis of microbiome composition between healthy and unhealthy human populations, and the metagenomic approaches with new powerful sequencing technologies, have enabled the screening of huge ecosystems, such as the human intestine microbiome [52,53]. The new strategy is called probiogenomics. The use of sequencing technologies combined with advanced computational methods allowed for better identification and characterization of probiotic microorganisms, including NGPs [54]. The ecosystem of the human gut microbiome and its diversity is covered by the open-access program that aims to collect and analyze the microbiological data from human oral and gut samples obtained from several diseased patients and healthy cohorts with a metagenomics strategy called the Human Gut Microbiome Atlas (HGMA). It “provides quantitative information of global shotgun metagenomics of human gut microbiome (i) function/phenotype information of the human gut-associated metagenomic species pan-genomes (MSP) and (ii) provide a global map of the human gut microbiome. Global shotgun metagenomics of normal human gut microbiome from 20 countries are presented with (i) species abundance, (ii) gene richness, and (iii) enterotypes. In addition, comprehensive analysis enabled the identification of species enriched/depleted in 23 different diseases with functions/phenotype of respective species, detailing the dysbiosis in gut microbiome composition” [55].

It is well known that the human microbiome is essential for human health and is responsible for maintaining the host’s physiological balance preventing pathogen colonization [56]. Alterations in microbiome composition enable the invasion of dangerous rivals, which leads to various acute and chronic diseases. On the other hand, appropriate modulations in the human microbiome can prevent or even support the treatment of diseases. This is a widely adopted concept of probiotics and can constitute a solid background for implementation into the market of other novel microorganisms, including NGPs [57].

Classical probiotic strains belong to the genera *Bacillus*, *Lactobacillus*, and *Bifidobacterium* [6]. The safety of probiotics’ long use is confirmed by GRAS or QPS status [58,59]. Hundreds of scientific reports confirm probiotics’ health benefits. In the meta-analysis of Ritchie and Romanuk (74 studies, 84 trials, 10,351 participants, and numerous probiotic preparations), the efficacy of probiotics in gastrointestinal diseases was confirmed [60]. Regardless of the tremendous amount of extensive research work, the situation of the NGPs is different, because, so far, they have never been used as food. Furthermore, there is a lack of regulation for NGPs in both the EU and the US. In 2012, the US Food and Drug Administration presented a novel category of bioactive microorganisms that overlaps NGPs. They introduced the definition of live biotherapeutic products (LBP): “a biological product that: (1) contains live organisms, such as bacteria; (2) is applicable to the prevention, treatment, or cure of a disease or condition of human beings; and (3) is not a vaccine” [61]. Their safety is not proven by a long history of safe use. The NGPs’ potentially positive effects on health are insufficient for authorities for their acceptance. Therefore, NGPs are not recognized as microorganisms intended for human use to reach the market as food, a food supplement, or a drug. In Europe, all microorganisms have to be evaluated by the EFSA before admission as a novel food component or as a drug. The European Pharmacopoeia, since 2019, includes a category of products that are intended to prevent or treat diseases as medicinal products and require registration according to the rules of newly developed drugs [62]. The authorities’ expectations and guidelines for LBP are not different from other biological products, and their development should comply with the requirements including proven safety and efficacy in the intended population [63]. Therefore, it seems that nowadays, for LBP, the only possible way for human-use registration is the path with the requirements intended for investigating new drugs according to the FDA rules in the US and the European Medicines Agency (EMA) in Europe. It means that, for NGPs, product approval, i.e., a series of expensive clinical trials with precise documentation (phases 1–3), should be conducted to establish safety, dose ranges, side effects, and benefits [3]. Despite NGPs registration difficulties due to their classifications as medicinal products, there is an increased research interest in a better understanding of the human microbiome’s role in maintaining a healthy balance. Therefore, scientific exploration should not only elucidate the mechanism and biological effects of NGPs but also provide scientific facts to establish their status for legal application with the intention to prevent or treat diseases.

The European Commission funded a collaborative project: MetaHIT (Metagenomics of the Human Intestinal Tract), which focuses on the constitution of a gene catalog of the intestinal microbiota and interactions of the microbiota with the human host. According to the official website: “the research undertaken in MetaHIT will lead to considerable progress in the understanding of the human metagenome, by the identification of functions of the gut microbiota. Discovery of associations between these functions and human health will open avenues for numerous diagnostic and therapeutic applications” [59]. Another great microbiome project is The Human Microbiome Project (HMP), supported by the National Institutes of Health (NIH). The mission was to create “resources that would enable the comprehensive characterization of the human microbiome and analysis of its role in human health and disease” [64].

The gut bacteria, which recently have been isolated, and evaluated as potentially beneficial for health, called NGPs belong to genera *Bacteroides, Clostridium, Faecalibacterium,* and *Akkermansia* [3]. There is a growing number of candidates for a potentially beneficial bacteria, the most promising are *Akkermansia muciniphila*, *Faecalibacterium prausnitzii*, *Anaerobutyricum hallii*, *Bacteroides* spp., *Roseburia* spp. and *Clostridium butyricum Bacteroides uniformis*, *Bacteroides acidifaciens*, *Bacteroides thetaiotaomicron*, *Prevotella copri*, *Christensenella minuta*, and *Parabacteroides goldsteinii* [65,66]. Table 1 shows the characteristic of selected candidates of next-generation probiotics.

The large-scale use of NGPs for future applications is challenging but promising. Many drawbacks, including sensitivity to oxygen and gastric conditions, and specific nutritional requirements, are necessary to be overcome in the development phase of commercial products [67]. Tailored strategy, practical manufacturing process, and appropriate storage are crucial for providing high numbers of viable cells in commercial products. The formula of NGPs delivery, i.e., precisely invented coating systems, should ensure the survival of bacteria in the digestive tract with the target delivery in the intestine [65,68]. More intense investigations are required to implement and open the door for NGPs in the food market as safe and efficient biotherapeutic products for human consumption.

### 4.1. Mechanism and Biological Effects of the Next Generation of Probiotics in the Balance of the Human Microbiome and in the Protection of Healthy Organisms from Dangerous Rivals

The role of next-generation probiotics in the microbiome is not fully clarified. Recent advances improved our knowledge, but it seems that, so far, we are at the beginning of the long path to elucidate unknown aspects and the mechanisms of how next-generation probiotics are associated with disease and health states. Gut microbiome composition is associated with host phenotypes and many factors, including the relation to diet, pharmaceutical intake as well as to metabolic disorders, disease risk factors, and disease development. It reflects also the host’s lifestyle and behaviors. The variety of host–microbiome interrelationships and co-varying factors require more research in the near future to better understand the mechanisms and correlations. With the background of many studies, we are convinced that the microbiome possesses great diagnostic potential and its appropriate manipulation gives an opportunity for targeted therapeutic intervention [66,69]. The relationship between the gut microbiome and immune system as well as bidirectional interplay with many host organs and systems, such as gastrointestinal, cardiovascular, cerebrovascular, pulmonary, endocrine, kidney, and skin, is relevant to health and disease [70,71].

The intestine microbiota can influence the host immune system mainly through the gut-associated lymphoid tissue as well as affect the host metabolism. The immunomodulatory effects resulting from dynamic changes in the vast spectrum of intestinal microorganisms are transmitted from gastrointestinal tract epithelial cells by toll-like receptors (TLRs) and nucleotide-binding oligomerization domain-containing protein (NOD)-like receptors (NLRs), which have an impact on innate and adaptive immunity [72]. The host physiology is maintained and regulated by microbiota metabolites that include folate, indoles, secondary bile acids, trimethylamine-N-oxide (TMAO), serotonin, gamma amino butyric acid (GABA), and also short chain fatty acids (SCFAs, acetate, propionate, and butyrate) [73]. SCFAs maintain metabolic and immunologic homeostasis, and intestine barrier integrity [74]. The most important SCFA is butyrate, which, along with propionate, constitutes an energy source for gastrointestinal cells and activates gluconeogenesis. Butyrate has an anticancerogenic profile and demonstrates anti-inflammatory effects [65]. Some NGPs (*Akkermansia muciniphila* and *Faecalibacterium prausnitzii*) play an important role in preserving and straightening the mucus intestinal barrier. They stimulate the production of bioactive compounds, i.e., tight-junction proteins, and can intensify mucus layer renewal. It reduces gut permeability, deteriorates inflammation, and is probably harmful to the proliferation of pathogenic bacteria [75]. SCFAs (mainly acetate, propionate, and butyrate) reduce the cholesterol level by inhibiting its synthesis, enhancing the conversion of cholesterol to secondary bile acids, inhibiting intestinal cholesterol absorption, increasing the efflux of cholesterol from macrophages, and stimulating leptin production, which decreases cholesterol synthesis in the liver. Another related to the microbiome mechanism is the transformation of cholesterol into coprostanol by the commensal bacteria, which lowers serum cholesterol level. Bacteria can also incorporate cholesterol into the cellular membrane [75]. The novel microbiota-derived mediator that can affect immune function, and have an impact on gut homeostasis and disease development are extracellular vesicles (EVs) [76]. One of the postulated pathophysiological mechanisms involved in the etiology of irritable bowel syndrome and inflammatory bowel disease is an imbalance of the mucosal serotonin system. The study of Yaghoubfar et al. (2021) indicated that NGPs *A. muciniphila* and *F. prausnitzii* and probiotic-derived EVs could modulate serotonin system-related genes. Hence, they could maintain the homeostasis of the gut serotonin system and straighten the intestinal mucosal barrier, and modulate immune functions [77].

### 4.2. The Next Generation of Probiotics in the Prevention and Treatment of Illnesses

The main target for the novel category of next-generation probiotics is gastrointestinal tract-related disorders, i.e., chronic gut inflammation, irritable bowel syndrome, ulcerative colitis, and Crohn’s disease. Moreover, gut dysbiosis in microbiota is able to affect more organs with the involvement of axes, such as the gut–lung, gut–brain, gut–heart, and gut–skin axes. The alterations within the gut microbiome interplay with a broad spectrum of diseases, such as metabolic disorders including diabetes, hyperlipidemia, and obesity, cancers of different organs, immune system disorders, chronic liver diseases, cardiovascular diseases including hypertension and coronary artery disease, neurological and psychiatric disorders, lung diseases including chronic obstructive pulmonary disease and even SARS-CoV-2, and skin diseases including autoimmunological allergies and psoriasis [70].

Inflammatory bowel disease (IBD) is characterized by microbiome dysbiosis. There is a reduction of SCFA-producing bacteria such as *Bacteroides*, *Eubacterium*, *Faecalibacterium*, and *Ruminococcus* and a rise in bacteria negative for host homeostasis such as *Proteobacteria* and *Actinobacteria*. An impaired intestinal homeostasis and lack of cytoprotective metabolites are responsible for intestinal barrier impairment and increased permeability [78,79].

The role of the gut microbiome in diabetes is not fully understood. It is known that microbiome composition changes may cause increased permeability of the intestinal barrier causing an uncontrolled entry of antigens into the blood. These antigens could activate autoimmunity processes leading to the damage of pancreatic β cells and contributing to the occurrence of diabetic complications. In diabetic patients, the diversity of gut microbiota is reduced [80]. In the study of Karlsson et al. (2013), it was shown that the abundance of *Roseburia intestinalis* and *Faecalibacterium prausnitzii* in patients with type II diabetes (T2D) is decreased [81,82]. The study performed by Qin et al. (2010) showed that in comparison with healthy patients the diabetic patients have a decreased abundance of *Roseburia intestinalis*, *Roseburia inulinivorans*, *Eubacterium rectale*, *Faecalibacterium prausnitzii*, and *Clostridiales* sp. SS3/4 [83]. Gurung et al. (2022), in the analysis of 42 human studies of T2D patients, confirmed microbial associations with diabetes [82]. They concluded that the genera of *Bifidobacterium*, *Bacteroides*, *Faecalibacterium*, *Akkermansia*, and *Roseburia* were negatively associated with T2D, while the genera of *Ruminococcus, Fusobacterium*, and *Blautia* were positively associated with T2D. Many studies showed that microbiome metabolism products, such as acetate, propionate, and butyrate, contribute to improved glucose homeostasis and metabolism [84]. NGPs can also have an impact on T2D pharmacological treatment. Commonly used metformin increases the prevalence of mucin-degrading *A. muciniphila* and SCFA-producing genera, such as *Megasphaera* and *Blauti*, which could lead to the renewal of the intestinal mucus layer [85]. The NGPs’ relevance for metabolic disorders such as obesity was confirmed in several studies. In obese subjects, a negative correlation of the number of *A. muciniphila* cells was reported. While in weight-loss patients the number of cells of *Faecalibacterium prausnitzii* and *A. muciniphila* increased [86,87]. In the double-blind, randomized study of 40 obese, insulin-resistant volunteers, the three months of oral administration of the pasteurized bacteria *A. muciniphila* (10^10^ CFU) resulted in reduced levels of the relevant blood markers of liver dysfunction and inflammation, improved insulin sensitivity, and slightly decreased body weight [88]. Perraudeau et al. (2020) conducted a randomized placebo-controlled trial with 76 patients with T2D. The oral WBF-011 preparate, which contains inulin, *Akkermansia muciniphila*, *Clostridium beijerinckii*, *Clostridium butyricum*, *Bifidobacterium infantis*, and *Anaerobutyricum hallii*, was tested during 12 weeks of study. It was concluded that WBF-011 is safe and improves postprandial glucose control, especially in patients on metformin monotherapy [89].

Several properties of NGPs are beneficial for patients with cancer [90]. Research results suggest that the microbiome is involved in the etiopathogenesis of colorectal cancers as well as in cancers in other organs such as the lungs. Bacterial metabolites, such as SCFA and polyphenols, are considered to have anticancer activity [91]. NGPs can suppress the growth of cancer. In a study by Ma et al. (2020), *F. prausnitzii* was shown to suppress the growth of breast cancer cells through the inhibition of the IL-6/STAT3 pathway [92]. It was also concluded that *Faecalibacterium* may be helpful for the prevention of breast cancer, and the reduction of *Faecalibacterium* promotes the development of breast cancer. In another study with a mice model of lung cancer, *A. muciniphila* was shown to enhance immune regulation and could augment the antitumor effect of cisplatin [93]. The complications of radiotherapy and chemotherapy in the form of gastrointestinal mucositis and diarrhea among patients with cancer are common. Restoration of gut microbiome homeostasis could have pivotal significance in the alleviation of these complications [94]. In the study of Lapiere et al. (2020), in mice it was shown that *F. prausnitzii* is effective in the prevention of the acute breakdown of the colonic epithelial barrier after radiotherapy [95].

Many studies suggest the beneficial effect of the microbiome on atherosclerosis development and progression [96]. The gut microorganisms can reduce the cholesterol serum level and contribute to the decrease of cardiovascular disease risk. One of the postulated mechanisms is the bacterial production of trimethylamine N-oxide (TMAO), which is associated with atherosclerosis [97]. An example of NGPs’ antiatherosclerosis effect is *Bacteroides dorei* D8, which was shown to convert absorbable cholesterol to nonabsorbable coprostanol [3].

Another of NGPs’ positive healthy effects on the cardiovascular system is related to blood pressure control [98]. It is well known that microbiome diversity in hypertensive patients as well in hypertension animal models is decreased [99]. The Cardia study, with a cohort of 529 young hypertensive patients, showed that *Akkermansia muciniphila* was negatively associated with blood pressure, whereas *Veillonella* aligned with individuals who were hypertensive. However, when the statistical analysis included obesity, none of the microbiota was associated with hypertension or blood pressure level, which indicates a demand for further studies [100].

The “leaky gut” hypothesis suggests that the increased permeability of the gut epithelial barrier is associated with neuropsychiatric disorders [101]. The activation of pro-inflammatory mediators can influence various brain functions and could have an impact on mental health [102]. The relationship is bidirectional with the involvement of gastrointestinal, immunological, biochemical, and neuro-endocrine systems [103]. In 2013, the definition of psychobiotic was introduced: “a live microorganism that, when ingested in adequate amounts, produces a health benefit in patients suffering from psychiatric illness” [104]. Many studies confirmed the potential benefits of probiotics in a vast spectrum of neuropsychiatric illnesses, i.e., autism, schizophrenia, and neurodegenerative disorders such as Alzheimer’s disease and Parkinson’s disease [103]. NGPs that are SCFA producers (especially butyrate) are promising in the prevention and treatment of psychiatric and mental neurodegenerative disorders [105]. The results of Liśkiewicz et al.’s (2021) study confirmed the impact of gut microbiota on intestinal barrier integrity with the influence of depressive symptoms [106].

The next-generation probiotics emerged as a new group of key significant microorganisms in the maintenance of gut homeostasis and with promising therapeutic possibilities. Unfortunately, despite many reports confirming the efficacy in preventing and treating illnesses, the perspective of its common use is limited, mainly due to restrictive rules. Precise strain identification, achievements in understanding the mechanisms, and individualized indication for human use can contribute to successfully passing through the difficult regulatory process.

**Table 1 microorganisms-11-01714-t001:** Characteristics of selected candidates of next-generation probiotics.

NGP Candidate	Characteristic	Potential Health Benefit	References
*Akkermansia * *muciniphila*	Gram-negative, anaerobic, 3–5% of the intestinal flora	Plays a role in the mucus layer renewal (immunomodulatory protein ‘Amuc_1100’ of the bacterial outer membrane). Reduction of gut permeability with the promotion of mucin production.	[107] [108] [109]
*Faecalibacterium prausnitzii*	Gram-positive, absolute anaerobic, 5% of the intestinal flora	Ferments glucose and produces SCFAs (butyrate, formic acid, and D lactate). Amelioration of inflammation by producing a microbial anti-inflammatory molecule. Reduction of proinflammatory cytokines. Correlation with the course of chronic heart failure. Production of mucin and tight-junction proteins.	[110] [111] [112] [113] [114]
*Bacteroides fragilis* without enterotoxin gene	Gram-negative, absolute anaerobic, 1% of the intestinal flora	Amelioration of inflammation by producing polysaccharide A.	[115] [116] [117]
*Roseburia* spp.	Gram-positive, absolute anaerobic, 3–15% of the intestinal flora	Produces butyric acid. Positive effects on several diseases (inflammatory bowel disease). Inhibits intestinal inflammation (increase of anti-inflammatory cytokines).	[118] [119] [120]
*Anaerobutyricum hallii*	Gram-positive, anaerobic, 2–3% of the intestinal flora	Formation of intestinal propionate. Formation of antimicrobial peptides, i.e., reuterin.	[121] [122]
*Christensenella minuta*	Gram-positive, anaerobic	Improvement of metabolic disorders and obesity (limits adiposity gain in the recipient mice).	[123]

## 5. Acetic Acid Bacteria

Acetic acid bacteria are common in the environment. These bacteria are isolated from plants and products made from them, including fermented foods (e.g., in the fermentation of cocoa beans) and beverages (e.g., vinegar, kombucha, lambic sour beer). There are three promising biotechnologically and health-promoting genus of AAB, namely, *Acetobacter*, *Gluconobacter*, and *Komagataeibacter*. They take part in the processes of the oxidative fermentation of food, where they produce many compounds with bioactive properties, including postbiotic ones, including glucuronic and gluconic acids, D-saccharic acid 1,4-lactone (DSL), as well as other organic acids, and vitamin C. Additionally, AAB are probably involved in the conversion and protective properties of phenolic compounds [124,125,126].

One of the key metabolites produced by AAB is glucuronic acid (GlcUA). It plays a role in detoxifying the liver and eliminating xenobiotics from the body, which makes it a substance that supports liver function. In addition to exogenous toxins, glucuronic acid is also involved in the endobiotic elimination of bilirubin, which is a product of the hemolytic breakdown of red blood cells. The effect of coupling the pigment with GlcUA molecules is a greater solubility of the complex in water, which allows its secretion with bile and removal through the digestive tract. The conjugation of phenols with GlcUA improves their transport and increases bioavailability. Enzymes such as glucuronosyltransferases (UGTs) and their isoform (e.g., UGT1A, UGT2B) found in the intestines are additionally involved in the glucuronidation of polyphenols. Glucuronic acid can also play a regulatory role, as it reduces the risk of steroid hormone deficiencies (e.g., estrogens, androgens, and glucocorticoids) by increasing their solubility in water, which improves their transport and bioavailability. At the same time, it prevents the accumulation of an excess of these hormones. The ability of glucuronides to influence the biological activity of endogenous estrogens after their deconjugation at the cellular level has been observed. In addition, they have a protective function in relation to polyunsaturated fatty acids (PUFA) by preventing lipid peroxidation, thanks to which PUFAs do not lose their health properties [127,128]. Furthermore, D-saccharin acid 1,4-lactone (DSL), a substance obtained in the course of transformations in the GlcUA pathway, has detoxifying and antioxidant properties. It allows for the easier removal of carcinogens, such as polycyclic aromatic hydrocarbons (PAHs), some nitrosamines or aromatic amines, and tumor promoters, which can be single steroid hormones and hepatotoxins [126]. Gluconic acid, formed by the oxidation of D-glucose, is another example of the metabolic activity of AAB. It improves the sensory properties of food products, giving a bitter but refreshing taste. It can be used as an additive and bio-preservative in the food industry.

AAB strains of the genus *Gluconobacter*, which have an oxidative capacity, can be used for the oxidative conversion of D-sorbitol to L-sorbose, which is an important intermediate in the production of L-ascorbic acid (vitamin C). Ascorbic acid plays an important role in human and animal nutrition and can be used as an antioxidant in the food industry [125,128].

In recent years, the subject of next-generation probiotics has become increasingly popular. They can be defined as microorganisms that meet the criteria of microorganisms with probiotic properties but have not yet performed their functions [129]. These limitations are caused, among others, by the FAO/WHO guidelines on the origin of potential probiotic microorganisms from the human digestive tract. On the other hand, an increasing number of studies provide information on obtaining new strains from health-promoting, probiotic properties from traditional, regional, and spontaneously fermented food [130,131]. The search for “new probiotics” is mainly limited to a group of lactic acid bacteria and selected fungi [132]. Less attention is paid to other groups of microorganisms that are safe for humans, known for years, which, in addition to technological usefulness, also have a health-promoting effect. A good example of new-generation probiotics or postbiotics are acetic acid bacteria. As aerobic bacteria, they do not demonstrate good survival during the passage through the digestive system, and even more so when colonizing the anaerobic environment of the large intestine. Their pro-health function should focus on post-biotic properties and the use of a number of bioactive metabolites. According to the definition of postbiotics formulated and promoted by ISAPP, postbiotics cannot be purified metabolites of microorganisms or have living cells be absent or negligible in the final product, and what is more, they should: (a) come from microorganisms, but they do not have to be a probiotic derivative; (b) be obtained by a deliberate process intended to break cell viability; (c) contain inactivated microbial cells and/or their metabolites or cellular components; and (d) have a proven health benefit to the host and a health safety rating [133].

A promising group of potential new-generation probiotics with a focus on their postbiotic effect are acetic acid bacteria. Despite the high potential and wide bioactive and application possibilities, the use of AAB on a mass scale is still limited and requires further research on this topic.

## 6. Factors Determining the Stability of Probiotics

Recently, due to the development of metagenomics and other genetic analysis, knowledge about the impact of metabolites produced by microorganisms with probiotic potential on the human body has been better understood. Probiotics are currently used as food ingredients as well as nutritional supplements due to their exceptional qualities and clinical significance.

The probiotics are available in a variety of medicinal dosage forms, including powders, tablets, lozenges, suspensions, pessaries, and more. Since the colon is the intended target site, the majority of them use oral administration techniques. Because they allow for the combination of two key differentiators into one form—higher potency and longer stability—capsules represent the most popular dosage form for probiotics [134].

There was also an increase in interest between the state of intestinal microbiota (its quantitative and qualitative composition) and the development of many civilization diseases, and therefore also the possibility of applying these microorganisms in “functional food” product development. Different functional foods, such as dairy products (milk, acidified milks, yogurts, cheeses, creams, and ice cream) and non-dairy items (meats and meat products, bread and cereal fiber snacks, juices, sorbets, and other fruit and vegetable products), are prepared using bacteria with probiotic properties [135,136].

While probiotics have gained popularity for their potential health benefits, there are also some technical challenges associated with their use. Probiotics are vulnerable to adverse effects during processing, storage, and passage through the gastrointestinal tract, thus reducing their viability [137].

### 6.1. Storage of Probiotic Preparations

Storage conditions for finished preparations in the form of mono- or subcultures of small particles with probiotic properties are crucial because storage conditions directly affect the biological viability and effectiveness of the preparation.

The important factors regarding the storage process include the composition of the probiotic preparation, temperature, oxygen content, storage time of water activity, and pH level.

Microorganisms with probiotic properties are highly sensitive to temperature, so they are usually stored in refrigerated conditions. In addition, to alleviate antioxidant stress in probiotic preparations, the addition of substances such as ascorbic acid and cysteine (different oxygen scavengers) is used [138,139].

### 6.2. Stability of Probiotics during the Processing

Probiotics that have been dehydrated have improved long-term viability and can be added to low-moisture food matrices, which also have good stability at both low and high temperatures. Probiotic survival, however, can be impacted by a number of elements related to the desiccation process, the physicochemical characteristics of the matrix, and the storage conditions [140,141,142].

Probiotics’ characteristics, including cell surface hydrophobicity, susceptibility to environmental stressors, and antibacterial activity, are impacted by drying [104]. The drying process and storing procedure have the biggest impact on the probiotic’s shelf life.

Microorganisms can be dried using a variety of techniques, including freeze drying, spray drying, vacuum drying, and fluidized beds [143]. However, a certain limitation on the use of these methods is the high cost.

### 6.3. Transport through the Gastrointestinal Tract

Probiotics need to survive the harsh conditions of the gastrointestinal tract and reach the target site in the gut alive.

Among the most difficult barriers to overcome when digesting food, there are low pH values in the stomach, digestive enzyme activity, as well as the presence of bile acid salts [144,145].

Factors that can protect the survival of bacteria are food ingredients, including prebiotics, i.e., non-viable nutrients that have a positive effect on the host’s health, given the modulation of the microorganism syndrome in the intestine [146].

Prebiotics are not digested by endogenous enzymes of the human gastrointestinal tract, but they end up intact in the colon, where they ferment, providing food for probiotic microorganisms. In addition, prebiotics are broken down by sucrose bacteria present in the lower gastrointestinal tract and have the ability to stimulate their growth. Inulin and oligofructose are the most effective and most commonly used prebiotics [147].

Encapsulation techniques, such as microencapsulation or enteric coating, can be used to enhance the survival of probiotics during transit through the digestive system. In the literature, several reports can be found reporting that cell encapsulation is a good solution due to improving the resistance of probiotic microorganisms to adverse conditions and, finally, their survival in many food matrices [148,149]. There are used, among others, such advanced techniques as emulsion [150,151], extrusion [152], spray chilling [153], spray drying [154], and fluidized beds [155].

The emulsion microencapsulation is an example of a potential technique for ensuring the safe delivery of next-generation probiotics applied to non-dairy products [156].

The viability of the probiotic organisms during storage product can be influenced by various variables, such as the food matrix used, the interactions with other current microbial species, the final acidity of the product, the water activity, the temperature, the availability of nutrients, growth stimulators and inhibitors, the inoculation level, the fermentation time, oxygen, and the processes used (lyophilization, spray drying, and freezing) [157].

Ensuring that a sufficient number of live and active probiotic cells reach the target site in the body can be a significant technical hurdle. The production of foods with probiotic claims is a challenge, especially due to the difficulties of survival and maintenance of the probiotic cells added to the foods under processing, storage, distribution, and consumption conditions. The functional probiotic food products should contain at least 10^6^–10^7^ cfu/g of probiotic bacteria at the end of its shelf life [158,159].

Currently, the potential for antibiotic-resistant genes for some bacterial strains is also a major threat associated with the use of probiotics in food, which, as a consequence, may lead to the transfer of antibiotic-resistant genes to pathogenic bacteria [160].

Current trends in probiotic food products have progressed away from dairy-based to plant-based products, beverages, and snacks. Probiotics are included in breakfast cereals, drinks, fruit snacks, chocolates, and confections in the intriguing non-dairy and non-fermented new-generation probiotic food products. Researchers have paid close attention to non-dairy probiotic food in recent years [161].

### 6.4. Innovative Solutions Affecting the Stability of Probiotics

Innovative solutions to increase the stability of probiotics include biofilms and lipid membranes [162,163,164].

A number of pathways that are controlled by extracellular matrix elements, connected to environmental factors, and influenced by bacterial community responses can lead to the production of biofilms [163].

The ability of biofilms to serve as both a physical adhesion enabler and a protective barrier has made them an innovative option for probiotic coatings. Bacterial biofilms have the potential to be used for gastrointestinal administration, as shown by the 125-fold increase in bioavailability of probiotics when *a Bacillus subtilis* biofilm is applied to their surface [164].

Probiotics may now be protected using quick and easy ways, which considerably protects their biological activity in harsh settings while enabling the creation of probiotic preparations in a fraction of the time.

In a previous study, *E. coli* and *Staphylococcus aureus* were vortexed with a solution of dimethylbenzoic acid and cholesterol to produce lipid membranes on the bacterial surface [162]. The bacteria’s intestinal bioavailability increased 4-fold, while their bioactivity remained unaltered after self-assembly, highlighting the great effectiveness of the bacteria coated with supramolecular self-assembly and the promise of this probiotic delivery technology.

It is interesting to dispel the idea that the application of edible nanomaterials, which has no toxic effect on the food matrix, improves the bioavailability of bioactive components in the food and enhances the compatibility of the components in the matrix [165].

Nanostructured particles are an alternative for creating tailored probiotic delivery systems because their pore size and diameter can be regulated to slow down their release rate in the harsh environment of the gastrointestinal tract [166].

## 7. Conclusions

Functional food using microorganisms with probiotic properties is a trend of the future. The current state of advancement of science and technology allows the production of food that has a positive impact on health while meeting the expectations of modern consumers. The concept of “beneficial microbes” includes probiotic, as well as potentially probiotic microorganisms (e.g., NGPs) and postbiotic producers, such as acetic acid bacteria. The positive effects of beneficial bacteria are reported in prophylaxis and the treatment of many common chronic and acute diseases of gastrointestinal, cardiovascular, neurological, pulmonary, metabolic, and immunological systems. Microorganisms with probiotic properties can also be isolated from so-called unconventional sources, such as from fermented food and, importantly, can be used in the production of novel functional products as starter cultures. The current trend in the science of human nutrition is undoubtedly the use of probiotics in terms of their beneficial effects on human health.

The homeostasis of the microbiome is essential for human health. Our knowledge of the still unexplored ecosystem is growing, and new information is gathered, opening new and fascinating possibilities relevant to health and disease. A broad spectrum of microorganisms beyond the classical probiotics, including NGPs and microorganisms from unconventional sources, have great potential for the treatment or prophylaxis of many health disorders. Generally, they are intended for use by healthy consumers as well as diseased patients. Nowadays, restrictive authorities’ regulations, especially regarding safety considerations, limit the introduction of live biotherapeutic products for human use. Health benefits and safety and a precisely identified target population need to be defined and endorsed in strain-specific double-blind randomized trials. It seems that, due to their specificity and uniqueness, NGPs are more suitable for use in the form of pharmaceuticals. However, we believe that in the future a new trend in their use will be associated with an increase in the number of their potential applications in food processing.

## Figures and Tables

**Figure 1 microorganisms-11-01714-f001:**
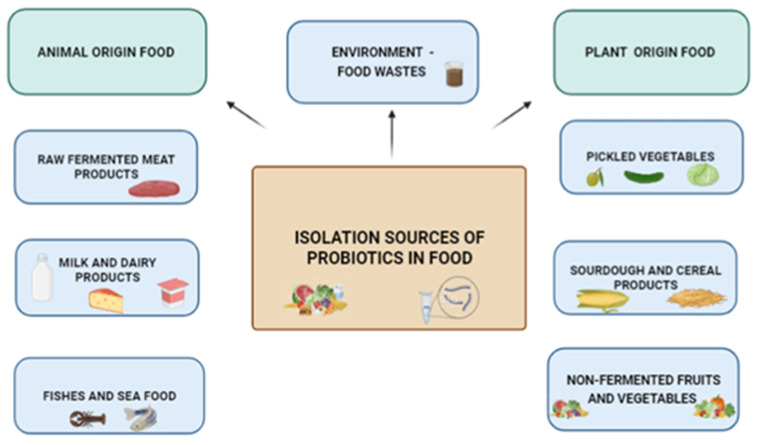
The isolation sources of probiotic microorganisms in food.

## Data Availability

Data are contained within the article.

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
