# Peer review of "Beneficial Bacteria Isolated from Food in Relation to the Next Generation of Probiotics"

_microorganisms, 2023, doi:10.3390/microorganisms11071714_

Round 1
Reviewer 1 Report
This review covered the current knowledge of some next-generation of probiotics (NGP) and acetic acid bacteria as new probiotic candidates.
The authors described that NGPs had been isolated from various foods, and their genetic diversity and many health benefits are being identified. In addition, the author suggests that NGPs would be included in a new category of probiotics if safety studies that can satisfy the authorities' restrictive rules are supplemented.
There are some minor comments.
1) What is the full name of UGT1A in line 260?
2) What is the full name of LBPs in line 399?
3) What is the species name of Clostridium in line 430?
4) The comma should be removed from "postprandial glucose, control" in line 545.
5) In the word "therosclerosis" in line 567, the initial "a" is missing.
Author Response
Manuscript ID: microorganisms-2454488
Response to Reviewer 1
We would like to thank the Reviewer for careful and thorough reading of this manuscript and for the thoughtful comments and constructive suggestions, which help to improve the quality of this article. All the changes in manuscript were marked in red color.
There are some minor comments.
- What is the full name of UGT1A in line 260?
Response: UGTs – a family of UDP-glucuronosyltransferases - enzymes are expressed at high levels in human liver. We added this information to the manuscript, lines 574-575: Enzymes glucuronosyltransferases (UGTs) and their isoform (e.g. UGT1A, UGT2B)
- What is the full name of LBPs in line 399?
Response: Full name was provided in red, line 330: live biotherapeutics products (LBP)
- What is the species name of Clostridium in line 430?
Response: Clostridium butyricum. Repeated Clostridium was deleted: line 370.
- The comma should be removed from "postprandial glucose, control" in line 545.
Response: The comma was removed, line 482.
5) In the word "therosclerosis" in line 567, the initial "a" is missing.
Response: According to the Reviewer "a" was added, line 504.
We attached revised manuscript.
We sincerely thank you for suggestions.
Best regards,
authors

Reviewer 2 Report
Some species or genus names of bacteria are not in italics.I suggest you check the manuscript for this.
Author Response
Manuscript ID: microorganisms-2454488
Response to Reviewer 2
We would like to thank the Reviewer for careful and thorough reading of this manuscript and for the thoughtful comments and constructive suggestions, which help to improve the quality of this article. All the changes in manuscript were marked in red color.
Comment: Some species or genus names of bacteria are not in italics. I suggest you check the manuscript for this.
Response: The manuscript was checked and species and genus names of bacteria were changed in italics.
We attached the revised manuscript.
We sincerely thank you for suggestions.
Best regards,
authors

Reviewer 3 Report
Your manuscript entitled " Beneficial bacteria isolated from food as the next generation of probiotics" targets a very interesting and at the same time complex topic. From my point of view, the topic is very meaningful for readers, especially considering the definition and source of next generation probiotics, but the manuscript needs to be strongly improved with major revisions to be suitable for publication in Microorganisms.
Specific comments on the content and on the structure of the manuscript:
1.Title: the title is not consistent with the content of this manuscript. In addition, the structure of this manuscript is very complex and confusing, making it very hard for readers to understand the content.
2.What’s the relationship between the next generation of probiotics and acetic acid bacteria? Why did the authors discuss them separately?
3.The authors aimed to provide an overall information of NGPs. However, a detailed introduction of NGPs were missed in the section 1.
4. Section 2.3: why did the authors only discuss the industrial use of lactic acid bacterial strains? The objective of this text is probiotics and probiotic microorganisms. I think this paragraph should provide a detailed discussion for these microorganisms.
5.Section 3: it is not clear why authors decided to separately discuss the acetic acid bacteria? There should be a clear logic for any content throughout the manuscript.
6. Section 5: Did the authors mean that the beneficial microorganisms form the ecosystem are the GNPs? The authors should state the concept of NGPs clearly.
7. The subtitle of section 6 should be revised. In addition, the stability of probiotics during the processing, storage, and transport through the gastrointestinal tract is crucial for the function of probiotics. Therefore, the emerging techniques used in the area of probiotic stabilization should be discussed in this part. I would also suggest the introduction of the following relevant references:
Elham, A., Jafari S M. Advances in Spray-Drying Encapsulation of Food Bioactive Ingredients: From Microcapsules to Nanocapsules [J]. Annual Review of Food Science & Technology, 2019, 10:103-131.
Feng, K., Huangfu, L L., Chuan, C D., et al. Electrospinning and Electrospraying: Emerging Techniques for Probiotic Stabilization and Application [J]. Polymers, 2023, 15(10):2402.
Callebe, C S., Verruck, S., Ambrosi, A., et al. Innovation and Trends in Probiotic Microencapsulation by Emulsification Techniques [J]. Food Engineering Reviews, 2022, 14(3):462-490.
Minor editing of English language required.
Author Response
Manuscript ID: microorganisms-2454488
Response to Reviewer 3:
We would like to thank the Reviewer for careful and thorough reading of this manuscript and for the thoughtful comments and constructive suggestions, which help to improve the quality of this article. All the changes in manuscript were marked in red color.
Your manuscript entitled " Beneficial bacteria isolated from food as the next generation of probiotics" targets a very interesting and at the same time complex topic. From my point of view, the topic is very meaningful for readers, especially considering the definition and source of next generation probiotics, but the manuscript needs to be strongly improved with major revisions to be suitable for publication in Microorganisms.
Specific comments on the content and on the structure of the manuscript:
Comment 1: Title: the title is not consistent with the content of this manuscript. In addition, the structure of this manuscript is very complex and confusing, making it very hard for readers to understand the content.
Response: According to the Reviewer’s suggestions some changes were made in the manuscript. All the changes were made in red. The structure of the manuscript was changed. The section 3 was moved to section 5.
The title of the manuscript was changed: Beneficial bacteria isolated from food in relation to the next generation of probiotics
Comment 2: What’s the relationship between the next generation of probiotics and acetic acid bacteria? Why did the authors discuss them separately?
Response: In our review we presented some “new probiotics”. The NGPS and acetic acid bacteria (AAB) were chosen because of their novelty and promising bioactive possibilities. There are many reports of the NGPs and AAB beneficial properties in shaping human health. Acetic acid bacteria are microorganisms that exhibit a number of health-promoting properties, including post-biotic ones. The authors suggest considering qualifying AAB as a future perspective in the development of the subject of probiotic or beneficial microorganisms. AAB meet the criteria of post-biotics. Due to the current state of knowledge about NGPs, the authors moved the section on AAB to the end of the article as a promising research direction in the field of beneficial microorganisms.
According to the some differences we decided to discuss NGPs and AAB in separate sections. AAB are isolated from so-called unconventional sources: plants, including fermented foods, while the source of the NGPs is human microbiome. Besides, it seems that acetic acid bacteria are more suitable for functional foods, while NGPs are more suitable for use in the form of pharmaceuticals.
Comment 3: The authors aimed to provide an overall information of NGPs. However, a detailed introduction of NGPs were missed in the section 1.
Response: According to the Reviewer’s suggestions short NGPs characteristic in the section 1 was added.
Comment 4: Section 2.3: why did the authors only discuss the industrial use of lactic acid bacterial strains? The objective of this text is probiotics and probiotic microorganisms. I think this paragraph should provide a detailed discussion for these microorganisms.
Response: According to the Reviewer’s suggestions the text was modified (text in red).
Comment 5:it is not clear why authors decided to separately discuss the acetic acid bacteria? There should be a clear logic for any content throughout the manuscript.
Response: Some explanation are in comment 2. Moreover AAB are isolated from fermented products constitute the microflora of the environment in which the products were produced. They may constitute an interesting alternative to gut bacteria. The structure of the article was changed. The subchapter has been moved to end of article (point 5) as a future perspective, which will facilitate the reception of the message about AAB as a well-developed research topic on beneficial microorganisms, like a probiotics and postbiotics.
Comment 6: Did the authors mean that the beneficial microorganisms form the ecosystem are the GNPs? The authors should state the concept of NGPs clearly.
Response: The Next generation of probiotic (NGPs) is a group of human derived microorganisms. To avoid confusion with term “beneficial bacteria”, “new probiotics” or “new generation of probiotics” some modifications were provided:
Abstract: lines 23 and 28
Introduction, lines 101-102 - text in red.
From the table 1 Glucunobacter oxydans was deleted.
Section 7, lines 747-749- text in red.
Comment 7: The subtitle of section 6 should be revised. In addition, the stability of probiotics during the processing, storage, and transport through the gastrointestinal tract is crucial for the function of probiotics. Therefore, the emerging techniques used in the area of probiotic stabilization should be discussed in this part. I would also suggest the introduction of the following relevant references:
- Elham, A., Jafari S M. Advances in Spray-Drying Encapsulation of Food Bioactive Ingredients: From Microcapsules to Nanocapsules [J]. Annual Review of Food Science & Technology, 2019, 10:103-131.
- Feng, K., Huangfu, L L., Chuan, C D., et al. Electrospinning and Electrospraying: Emerging Techniques for Probiotic Stabilization and Application [J]. Polymers, 2023, 15(10):2402.
- Callebe, C S., Verruck, S., Ambrosi, A., et al. Innovation and Trends in Probiotic Microencapsulation by Emulsification Techniques [J]. Food Engineering Reviews, 2022, 14(3):462-490.
Response: We sincerely thank you for this suggestion. The corrections have been done. The title and the text of the section no. 6 were revised. Also new references were completed: lines 646, 670, 696.
We attached the revised manuscript.
Best regards,
authors

Round 2
Reviewer 3 Report
The authors have addresed all the suggestions I mentioned. The manuscript can be accepted in present form.